# HYBRID DEFENSE STRATEGY AGAINST FACE RECOGNITION MODEL INVERSION ATTACK

## ABSTRACT

The utilization of personal sensitive data in training face recognition (FR) models poses significant privacy concerns, as adversaries can employ model inversion attacks (MIA) to infer the original training data. Existing defense methods, such as data augmentation and differential privacy, have been employed to mitigate this issue. However, these methods often fail to strike an optimal balance between privacy and accuracy. To address this limitation, this paper introduces an adaptive hybrid masking algorithm against MIA. Specifically, face images are masked in the frequency domain using an adaptive MixUp strategy. Unlike the traditional MixUp algorithm, which is predominantly used for data augmentation, our modified approach incorporates frequency domain mixing. Previous studies have shown that increasing the number of images mixed in MixUp can enhance privacy preservation but at the expense of reduced face recognition accuracy. To overcome this trade-off, we develop an enhanced adaptive MixUp strategy based on reinforcement learning, which enables us to mix a larger number of images while maintaining satisfactory recognition accuracy. To optimize privacy protection, we propose maximizing the reward function (i.e., the loss function of the FR system) during the training of the strategy network. While the loss function of the FR network is minimized in the phase of training the FR network. The strategy network and the face recognition network can be viewed as antagonistic entities in the training process, ultimately reaching a more balanced trade-off. Experimental results demonstrate that our proposed hybrid masking scheme outperforms existing defense algorithms in terms of privacy preservation and recognition accuracy against MIA.

## 1 INTRODUCTION

Face recognition (FR) has found wide applications in various practical systems, as face images provide unique identity information. If the face information is maliciously stolen, personal privacy will be compromised. Two privacy protection scenarios are listed here. The first is that private data needs to be provided to untrusted third parties for model training. To protect the privacy of data (face visualization information), data needs to be masked. The second is that the private data owner also carries out model training. The trained model will be provided to an untrusted third party for use. It is necessary to avoid inferring the original training data from the model.

For the first scenario, commonly used privacy protection techniques include differential privacy (DP) Zhao et al. (2020); Girgis et al. (2021) and data masking Huang et al. (2020); Wang et al. (2022b). DP protects privacy by adding noise to input data, gradients, and labels, which normally leads to a significant decline in task performance. Compared to DP, the masking method has no significant impact on the performance of the task Wang et al. (2022b). Typical data masking methods include MixUp Zhang et al. (2017), Instahide Huang et al. (2020) and PPFR-FD Wang et al. (2022b). Note that MixUp can also be used for data enhancement. In this case, the mixed data is utilized together with its original version in training. When MixUp is explored for masking, only the mixed data would be used in training. Unfortunately, both MixUp and Instahide have been successfully hacked (see Carlini et al. (2020)). Although MixUp and Instahide are hacked in the first scenario, they can be used in the second scenario for defending against the MIAs. Note that the MIA method based on generative adversarial networks (GAN) poses a threat to the defense methods in the second scenario.

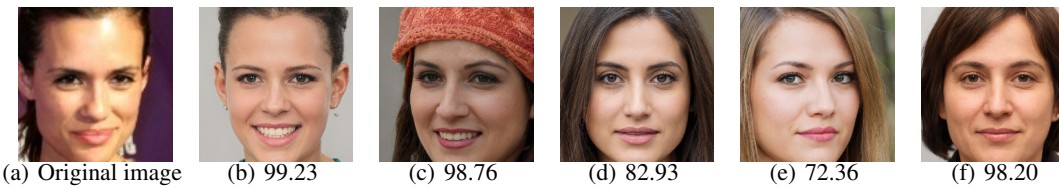

(a) Original image     (b) 99.23     (c) 98.76     (d) 82.93     (e) 72.36     (f) 98.20

Figure 1: Comparison of the results of MIA Zhang et al. (2020) and accuracy for different defense methods. From the left to right, we show the original image, the result of no defense, Mixup, Instahide DP and our denfense methods. The value below each image indicates recognition accuracy of the corresponding model. We expect to obtain a high-accuracy performance model while its result of MIA looks as much as different from the original image. Some existing defense methods cannot achieve a better trade-off between privacy and accuracy.

Moreover, the current methods always resist MIA at the expense of the performance decline of the model, and fail to achieve a good trade-off between privacy and accuracy (see Fig. 1).

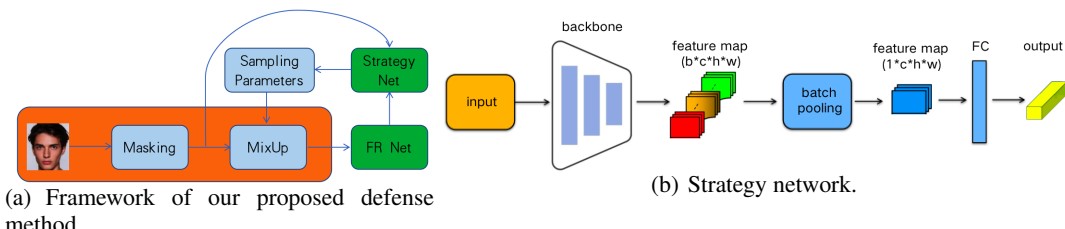

(a) Framework of our proposed defense method.

(b) Strategy network.

Figure 2: The framework of (a) our proposed defense method and (b) the strategy network.

This paper proposes a hybrid defense algorithm for MIA. In Fig. 2(a), it can be seen that after the original face image is masked by the data masking algorithm (here, we use PPFR-FD as the first masking step), we mix the outcome of PPFR-FD with MixUp. We expected to use the idea of hybrid masking to enhance privacy protection, but it was found that if the number of mixed masked images $k$ in MixUp is bigger than 3, the recognition performance would be significantly reduced (see Table 2). We thus use reinforcement learning (RL) Williams (1992) to adaptively select the parameter $k$. The more images mixed in the MixUp or Instaide algorithm, the stronger the privacy protection ability Huang et al. (2020). However, the more mixed images, the lower the recognition accuracy and the greater the value of the loss function. To enhance the ability of privacy protection, the strategy network in reinforcement learning tends to choose a larger mixed value. We take the loss function of the FR network as the reward function. For enhancing the ability of privacy protection, we maximize the reward function (i.e. the loss function of FR) when training the strategy network. The loss function of FR is minimized during the FR network training. The strategy and face recognition networks can be regarded as two opposing sides in the training process. Finally, the two reached a better trade-off between privacy and utility. Experiments also show the proposed method not only provides good privacy protection but also makes the accuracy comparable to that of the case when original face images are used during training.

Our contributions are as follows:

- An adaptive hybrid defense algorithm for MIA is proposed. The face image is masked by combining frequency-domain masking and RL-enhanced MixUp. Unlike conventional MixUp, we use MixUp in the frequency domain. Our proposed MixUp method based on RL can improve the privacy protection capability while retaining good recognition.

- For face image masking algorithms, we put forward a set of systematical measures to quantify the effect of masking and privacy protection. Experimental results show that the proposed adaptive hybrid defense algorithm has a better privacy-preserving ability and provides satisfactory recognition accuracy.

## 2 RELATED WORK

### 2.1 MODEL INVERSION ATTACKS

As we know, most MIAs utilize the ability generative generative adversarial networks (GANs) Goodfellow et al. (2020), and use images as the prior information to generate realistic images. Some MIAs avoided distributional shifts and relied on GANs trained on the same data distribution as the target model Zhang et al. (2020); Chen et al. (2021); Wang et al. (2021), used additional input information such as blurred pictures of a person Zhang et al. (2020), and tailored the attack and its image prior to specific target models Chen et al. (2021); Wang et al. (2021), restricting the reuse and flexibility of the attacks. And all approaches focused on low-resolution images, which limits the quality of the extracted features, and have yet to show their applicability for higher resolutions. Struppek et al. (2022) presents Plug & Play Attacks, which relax the dependency between the target model and image prior, and enable the use of a single GAN to attack a wide range of targets, requiring only minor adjustments to the attack. It also provides a more formal introduction to MIAs and theoretical consideration of ideal MIAs and possible degradation factors.

### 2.2 MASKING BASED DEFENSE METHODS

Common defense methods include data augmentation, differential privacy and other making methods. In Chamikara et al. (2020), privacy protection using the EigEnface Perturbation (PEEP) method is presented. It perturbates eigenfaces using DP Gong et al. (2020) and only stores perturbed data at the third-party servers that run the eigenface algorithm. But the FR accuracy of PEEP is much worse than that using original face images. Typical masking techniques include MixUp, which is originally a data enhancement method, Instahide and PPFR-FD. The first two methods have been recently hacked Carlini et al. (2020). The last method exposes some rough face information after black-box reconstruction. As a result, there are still some security risks. The common means to measure the defense method is to carry out MIA using SoTA methods. It can be seen from the experiments that this attack is difficult to infer the training data for the hacked masking algorithm (e.g., MixUp, Instahide). If the masking-based defense algorithm is used as a method to resist MIA, the more reasonable evaluation is to use the metrics of the masking algorithms, which makes the requirements for the masking-based defense methods more stringent. This paper adopts these stricter metrics.

In general, the masking effect is measured using the attack methods, which include white-box and black-box attacks. The white-box attack is customized because it assumes that the specific operation details of masking are known, and its attack method is also combined with specific masking steps. The black-box attack assumes that the attack model can be accessed multiple times and multiple input-output pairs can be obtained. In literature such as Huang et al. (2020); Wang et al. (2022b), the black-box attack is usually realized by GAN. Most works judge whether they can resist black-box attacks according to the visualization effect of reconstructed images from GAN, which lacks quantitative measure standards. Some literature Tanaka (2018); Wang et al. (2022b) uses the size of brute force search space, which is often used in cryptography, to quantify the ability of privacy protection at the pixel level. Due to the strong semantic correlation between image pixels, in most cases, sensitive information can be obtained without completely restoring all pixels. So it is inappropriate to use brute force search space to measure the privacy-preserving ability at the pixel level.

## 3 THREAT MODEL

We consider this scenario: the private data owner conducts model training, and the trained model will be provided to an untrusted third party for use. The untrusted third party can obtain the structure of the model and all parameter information. At the same time, we give an untrusted third party the ability to infer all masked data. This is a powerful assumption for attackers. The attacker attempts to deduce the original training data from the model and masked data.

## 4 METHODOLOGY

### 4.1 ADAPTIVE HYBRID MASKING

We provide detailed introductions of masking in the frequency domain as preliminary preparations in Appendix.

In MixUp, the mixed images used in this work come from the same batch. The processing steps are as follows. When the number of images participating in each mixing is set to $k$, $k$ batches are first obtained by randomly shuffling the images in the original batch $k-1$ times. For the convenience of explanation, assume that a supermatrix with the size of $batch * k$ is obtained, each *entry* in the supermatrix corresponds to an image, and the first column collects the images of the original batch. Next, we randomly generate the weight coefficient matrix of size $batch * k$ and normalize the rows so that the sum of the coefficients of each row is 1, and the maximum coefficient cannot exceed the specified value of 0.55. Then, each row of the image supermatrix is weighted with each coefficient matrix along the row direction, and finally, a $batch * 1$ mixed image supermatrix is obtained. Labels are mixed in a similar way. The difference is that labels are represented by weighted one-hot vectors, that is, a mixed image corresponds to a weighted one-hot vector, in which the non-zero value is no longer 1, but the previously randomly generated weighting coefficients.

Table 1: Comparison of face recognition accuracy between MixUp and Instahide in the RGB and frequency domains. The experiment set is the same as that in Section 5.1.

| Method | RGB domain (LFW) | Frequency domain (LFW) |
|---|---|---|
| MixUp($k$=2) | 86.38 | 98.76 |
| Instahide($k$=2) | 61.75 | 82.93 |

Note that the difference between Instahide and MixUp is that the former randomizes element signs based on the latter. Through the comparison between Instahide and MixUp in Table 1, we find that the accuracies of these two algorithms in the frequency domain are much better than that in the RGB domain. Moreover, the face recognition accuracy of MixUp is better than that of Instahide. Given this, we use frequency-domain MixUp in the following fusion method. The pseudo-code of the MixUp-based ArcFace algorithm is provided in Appendix Algorithm 1.

MixUp has two hyper-parameters. One is the maximum weight for the mixed images (the sum of all weights needs to be 1). Both MixUp and Instahide set this hyper-parameter to 0.65 by default. In this work, we choose a smaller value of 0.55 so that different images can contribute more to the mixed data, which leads to higher privacy protection capability. The other hyper-parameter is the number of images used in the mixing operation, denoted by $k$. It can be expected by intuition that a larger $k$ should result in higher privacy protection level ability at the cost of lower face recognition accuracy. To obtain a better trade-off between privacy protection and system utility, we propose an adaptive hybrid masking method based on RL. The basic idea is to sample the possible strategies using a strategy network during training.

As shown in Fig. 2(b), we select ResNet18 as the strategy network backbone, and its input is the PPFR-FD-masked version of the entire batch.The strategy network aims at generating the probability mass function (PMF) for the possible values of $k$, which are collected in the specified set, e.g., {2,3,4,5,6}. The softmax operation is introduced *after* the full-connection layer of ResNet18 to achieve this purpose. The above design enables the RL module to select a strategy, which is the number of masked images used in MixUp, by sampling $k$ from {2,3,4,5,6} according to the PMF estimated by the strategy network. Another modification we include is the average pooling operation before the full connection layer. This effectively creates a batch size of 1, which matches well with the need for generating a single PMF as output.

#### 4.1.1 LOSS FUNCTION

For participant $i$, when we don't use the adaptive MixUp module, i.e., PPFR-FD+MixUp with fixed $k$ (no strategy network exists), training the face recognition network attempts to solve

$$\min_{W_i} E_{(x_t,y_t)\sim D} L\left(FRNet\left(x_{mix}, W_i\right), y_t\right),$$ (1)

where $x_t$ and $y_t$ are the raw data and label in $x_{bt}$ and $y_{bt}$, $x_{bt}$ and $y_{bt}$ denote the raw data and label sets of a batch selected from the training set $D$ during the $t$th-round training. $L$ is the cross-entropy loss function. We use Arcface as the metric function for the face recognition network (FRNet), and the weights of FRNet are included in $W_i$. $x_{mix}$ is the output of MixUp($x_{bt}$,$k$) that mixes every image in $x_{bt}$ with other $k$ images in it. When the adaptive MixUp module is active, i.e., PPFR-FD+AdaMixUp with adaptive $k$ sampling according to the strategy network outcome, our goal is to protect privacy as much as possible, and the strategy network should choose the strategy that increases the loss of the FRNet. So training the face recognition network becomes solving

$$\min_{W_i} E_{(x_t,y_t)\sim D}\left(\max_{\phi_i} E_{k\sim p(k|x_{mt},\phi_i)} L\left(FRNet\left(x_{mix}, W_i\right), y_t\right)\right),\qquad(2)$$

where $\phi_i$ is the weight of the strategy network, $x_{mt}$ is the masked data by the masking algorithm. We use the strategy network to produce the sample-dependent and learnable strategy $p\left(k|x_{mt},\phi_i\right)$.

To train the strategy network, we first produce a batch of PPFR-FD-masked face images. The strategy network then outputs the associated PMF for $k$, according to which a sample of $k$ is drawn. The selected value of $k$ is used in MixUp to mix masked images in the batch. The mixed data $x_{mix}$ are input into the face recognition network for recognition and calculating the recognition loss (called reward1) using

$$L_1\left(\phi_i\right) = L\left(FRNet\left(x_{mix}, W_i\right), y_t\right).\qquad(3)$$

Based on $L_1\left(\phi_i\right)$, we train the face recognition network to obtain its updated version FRNet1. This newly trained face recognition network is used to calculate the loss of the strategy network only, and it is discarded after the current strategy network training epoch. The MixUp outcome $x_{mix}$ is fed to calculate the recognition loss (called reward2) using

$$L_2\left(\phi_i\right) = L\left(FRNet1\left(x_{mix}, \mathring{W}_i\right), y_t\right),\qquad(4)$$

where $\mathring{W}_i$ and FRNet1 are the updated weight and model FRNet after one-step training.

Combining the two loss terms in (3) and (4), as well as the loss term of FRNet itself, we transform the optimization problem in (2) into

$$\min_{W_i} E_{(x_t,y_t)\sim D}$$
$$\left(\max_{\phi_i} E_{k\sim p(k|x_{mt},\phi_i)}\left(L_0\left(W_i\right) + a\cdot L_1\left(\phi_i\right) + b\cdot L_2\left(\phi_i\right)\right)\right),\qquad(5)$$

where $L_0\left(W_i\right)$ is in fact $L\left(FRNet\left(x_{mix}, W_i\right), y_t\right)$ in (1), $a$ and $b$ are hyper-parameters with default values $a = 1$ and $b = 1$. In order to enhance the ability of privacy protection, the strategy network in reinforcement learning tends to choose a larger mixed value. We take the loss function of the FR network as the reward function. For enhancing the ability of privacy protection, we maximize the reward function (i.e. the loss function of FR) when training the strategy network. The loss function of FR is minimized during the FR network training. The strategy and face recognition networks can be regarded as two opposing sides in the training process.

### 4.1.2 SOLVING (5)

We propose an algorithm to solve (5) efficiently. The developed technique is based on alternative projection. Specifically, given the strategy network $\phi_i$, (5) reduces to training the face recognition network through solving

$$\min_{W_i} E_{(x_t,y_t)\sim D} E_{k\sim p(k|x_{mt},\phi_i)}\left(L_0\left(W_i\right)\right).\qquad(6)$$

Next, fixing the face recognition network $W_i$, (5) reduces to training the strategy network through solving

$$\max_{\phi_i} E_{(x_t,y_t)\sim D} E_{k\sim p(k|x_{mt},\phi_i)}\left(a\cdot L_1\left(\phi_i\right) + b\cdot L_2\left(\phi_i\right)\right).\qquad(7)$$

Note that the selection strategy $k$ can only take values in a finite discrete set. Following by the REINFORCE algorithm Williams (1992), we can transform (7) into

$$\max_{\phi_i} E_{(x_t,y_t)\sim D} E_{k\sim p(k|x_{mt},\phi_i)}\left(a\cdot L_1\left(\phi_i\right) + b\cdot L_2\left(\phi_i\right)\right)$$
$$= \max_{\phi_i} \sum_k \sum^{N_D} p\left(k|x_{mt},\phi_i\right)\left(a\cdot L_1\left(\phi_i\right) + b\cdot L_2\left(\phi_i\right)\right),\qquad(8)$$

where $N_D$ is the number of samples in the training batch.

It can be seen that within the proposed adaptive hybrid masking algorithm, two masking techniques, namely PPFR-FD and MixUp (with the adaptive selection of the number of images for mixing), are combined. The use of MixUp thus brings in extra privacy protection ability. Moreover, the introduction of the RL module (i.e., the strategy network) allows MixUp to mix more than three images wherever possible to improve privacy protection without greatly sacrificing FR accuracy. See Section 'Experiments' for experimental verification.

## 4.2 PERFORMANCE MEASURE FOR MASKING ALGORITHMS

If the masking-based defense algorithm is used as a method to resist MIA, the more reasonable evaluation is to use the metrics of the masking algorithms, which makes the requirements for the masking-based defense methods more stringent. This paper adopts these stricter metrics. Note there are no standard measures for quantifying the ability of privacy protection and impact on the recognition accuracy of a face masking algorithm. In this subsection, we advocate a set of performance measures for masking algorithms that consider their masking effect, attack effect and impact on recognition accuracy.

### 4.2.1 MASKING EFFECT

We evaluate the masking effect from the perspective of the visual effect and feature-level evaluation. **Visual effect**: The metrics for evaluating image quality generally include learned perceptual image patch similarity (LPIPS), peak signal-to-noise ratio (PSNR) and structural similarity (SSIM). Zhang et al. (2018) proves that using LPIPS is more effective than other metrics. Here, we also use LPIPS as a performance measure. During the evaluation process, LPIPS for the original face image and the masked image are calculated. We average the result over the whole dataset, and the output is denoted as the $S1$ index. The larger $S1$ is, the greater the difference between the original image and the masked image would be at the visual level, implying a better masking effect.
**Feature level evaluation**: During the evaluation process, a model $F1$ trained with the original face images is provided to extract features. The original face image and the masking image are respectively input into the model to obtain the corresponding feature vectors. The cosine similarity of the two feature vectors is calculated as the score for a particular image. Conducting the same operation on all the samples, the mean score is found and subtracted from one to produce the $S2$ index. A larger $S2$ indicates lower similarity between the original image and masked image at the feature level, thus meaning a better masking effect.

### 4.2.2 ATTACK EFFECT

The attack methods can be generally divided into white-box attacks and black-box attacks. White-box attack methods assume the knowledge of all the details of privacy protection protocols in advance. Generally, their attack methods are customized, that is, different privacy protection methods may be associated with different white-box attacks. To emphasize the generality of the evaluation criteria, we use Conditional GAN (CGAN) as the basis of the attack method. The masked data is obtained by passing the auxiliary dataset through the masking algorithm, which produces pairs of original data and masked data. Masked data and noise are used as input and original images as output. CGAN which has a strong fitting ability is used to model the corresponding relationship between these data pairs. However, if we know the details of privacy protection methods in advance, we can also integrate them into the reconstruction process as *a priori* information. Note that when benchmarking multiple masking algorithms at the same time, we need to train CGAN using the same network structure and training dataset, and then generate the recovered images based on the masked test data and trained CGAN. Again, we evaluate the attack effect from the perspective of the visual effect and feature-level evaluation using recovered images. As the methods in the previous section, we use the $S3$ and $S4$ indexes to denote visual effect and feature-level evaluation.

### 4.2.3 RECOGNITION ACCURACY

We use the trained model $F1$ in the previous section (or retrain a network) to evaluate the recognition accuracy for the original test data, and the obtained recognition accuracy is denoted as $acc_{bsl}$. Then, we train another face recognition network $F2$ with masked data and test on the masked test set to

obtain the recognition accuracy $acc_{mask}$. $acc_{mask}/acc_{bsl}$ is the recognition accuracy indicator, i.e., $Score_{cls}$. The larger $Score_{cls}$ is, the less impact of the masking effect on the recognition accuracy is.

### 4.2.4 COMPREHENSIVE EVALUATION

We fuse indicators using different weights to produce the privacy score $Score_{pp}$ and recognition accuracy score $Score_{cls}$.

$$Score_{pp} = Normalization(\alpha(s1 + s2) + \beta(s3 + s4)), \tag{9}$$

$$Score_{cls} = Normalization(acc_{mask}), \tag{10}$$

where $\alpha$ and $\beta$ are the weights for measures at the visual level and attack level, and $Normalization(*)$ stands for normalization operation. Usually, masking images at the visual level is easier to achieve, so we will pay more attention to the indicators at the attack level. With this observation in mind, $\alpha$ and $\beta$ are set to be 0.4 and 0.6 in the experiments. In the experiments, the normalized range is [0.5, 1].

Finally, taking the product of the above two scores yields the following scalar comprehensive score to measure the overall privacy protection ability and recognition accuracy

$$Score = (Score_{pp} + Score_{cls})/2. \tag{11}$$

## 5 EXPERIMENTS

We assume that malicious server attackers, utilizing shared gradients and weights from clients, have sufficient ability to reconstruct the data participating in the process of network training. This worst-case assumption is necessary to measure the privacy protection ability of the masking algorithms. Note that the reason why the federal face recognition algorithms introduced in Section 2.1.2 is not added in the comparative experiments is that these algorithms are difficult to resist the malicious attack methods set in Section 3. However, we can better resist the malicious attack method by adding our proposed masked algorithm based on these algorithms (see experiments in Section 5.2).

### 5.1 EXPERIMENTS IN CENTRALIZED SCENARIOS

Table 2: Comparison of the face recognition accuracy among methods with and without privacy protection for different datasets in centralized scenarios.

| Method | Mask | $Score$ | $Score_{pp}$ | $Score_{cls}$(LFW) | LFW | CFP | AgeDB | CALFW |
|--------|------|---------|--------------|--------------------|-----|-----|-------|-------|
| ArcFace Deng et al. (2019) | No | - | - | - | 99.23 | 96.26 | 95.53 | 94.92 |
| Masking+AdaMixUp(k:2-6) | Yes | 0.968 | 0.979 | 0.957 | 96.15 | 93.63 | 94.73 | 91.20 |
| **Masking+AdaMixUp(k:2-5)** | Yes | **0.981** | 0.975 | 0.987 | 98.20 | 95.29 | 95.11 | 93.25 |
| Masking+AdaMixUp(k:2-4) | Yes | 0.959 | 0.925 | 0.992 | 98.43 | 95.33 | 95.18 | 93.71 |
| Masking+MixUp(k=2) | Yes | 0.912 | 0.825 | 0.999 | 98.92 | 95.29 | 95.66 | 94.03 |
| Masking+MixUp(k=3) | Yes | 0.940 | 0.892 | 0.989 | 98.20 | 94.81 | 95.03 | 93.38 |
| Masking+MixUp(k=4) | Yes | 0.877 | 0.975 | 0.779 | 84.62 | 80.07 | 80.62 | 77.90 |
| Masking+MixUp(k=5) | Yes | 0.796 | 0.988 | 0.605 | 73.31 | 69.51 | 68.55 | 66.49 |
| Masking+MixUp(k=6) | Yes | 0.750 | 1.000 | 0.500 | 66.53 | 63.35 | 65.83 | 63.07 |
| MixUp-FD(k=2)Zhang et al. (2017) | Yes | 0.749 | 0.500 | 0.997 | 98.76 | 95.22 | 94.92 | 93.88 |
| PPFR-FD(Masking)Wang et al. (2022b) | Yes | 0.806 | 0.613 | 1.000 | 98.93 | 95.54 | 95.08 | 93.65 |
| DP Chamikara et al. (2020) | Yes | 0.753 | 0.917 | 0.590 | 72.36 | 69.71 | 68.54 | 66.80 |
| InstaHide-FD(k=2)Huang et al. (2020) | Yes | 0.733 | 0.713 | 0.753 | 82.93 | 77.01 | 76.76 | 73.13 |

Table 3: Average proportion of each candidate value $k$ for Masking+AdaMixUp when its candidate value sets are {2,3,4,5,6}, {2,3,4,5}, {2,3,4}, respectively.

| Method | Average proportion |
|--------|--------------------|
| Masking+AdaMixUp(k:2-6) | [0.17,0.25,0.13,0.26,0.19] |
| Masking+AdaMixUp(k:2-5) | [0.21,0.28,0.16,0.35,-] |
| Masking+AdaMixUp(k:2-4) | [0.31,0.44,0.25,-,-] |

We use CASIA Yi et al. (2014) as the training set. 4 benchmarks including LFW Zhang & Deng (2016), CFP-FP Sengupta et al. (2016), AgeDB Moschoglou et al. (2017), CALFW Zhang & Deng (2016)are used to evaluate the performance of different algorithms following the standard evaluation protocols. We train the baseline model on ResNet50 He et al. (2016) backbone with SE-blocks Hu et al. (2020) and batch size of 512 using the metric and loss function similar to ArcFace. The head of the baseline model is BackBone-Flatten-FC-BN with embedding dimensions of 512 and the dropout probability of 0.4 to output the embedding feature. All models are trained for 50 epochs using the SGD optimizer with the momentum of 0.9, weight decay of 0.0001.

We first investigate the centralized training, that is, all the data are in one client. Compared methods include Arcface (using original data) and the existing masking algorithms PPFR-FD, DP, Instahide and MixUp. Instahide-Fd and MixUp-FD in Table 2 denote Instahide and MixUp using the frequency domain data. As shown in Table 1, the performance of Instahide and MixUp is very poor for the original images in the RGB domain. In DP Chamikara et al. (2020), $\epsilon$ is set to 3 and laplacian noise is introduced. For InstaHide, we set $k$ to 2, that is, only two images in the same data batch used for encryption. To compare the effects of the adaptive module, we provide experiments of a combination of PPFR-FD and MixUp without the introduction of adaptive modules, i.e., Masking+MixUp when $k$=2, 3, 4, 5 and 6, respectively. We also investigate experiments of a combination of PPFR-FD and adaptive MixUp, i.e., AdaMasking, which considers different candidate value sets ({2,3,4}, {2,3,4,5} and {2,3,4,5,6}). As shown in Table 2, when $k$=4, the performance of MixUp used alone has decreased significantly. Table 3 provides the average proportion of each candidate value $k$ for Masking+AdaMixUp when its candidate value sets are {2,3,4,5,6}, {2,3,4,5}, {2,3,4}, respectively. For the case of AdaMasking, when $k$ belongs to {2,3,4}, the proportion of $k$=4 in the whole training process is about 30%. while the recognition accuracy is only 0.5% lower than that of MixUp ($k$=2). When $k$ belongs to {2,3,4,5}, the proportion of $k > 3$ in the whole training process is about 50%, while the recognition accuracy is only 0.82% lower than that of MixUp ($k$=2). When $k$ belongs to {2,3,4,5,6}, the proportion of $k > 3$ in the whole training process is about 60%, while the recognition accuracy is 2.77% lower than that of MixUp ($k$=2). Since DP directly adds noise to the image or gradient, the recognition performance is significantly degraded. Combined with the privacy scores, we can see that although the recognition performance of PPFR-FD, Instahide and MixUp ($k$=2 or 3) is good, the privacy score is not as good as that of AdaMasking.

The reconstruction results are shown in Fig. 3. Visual sense from Fig. 3 and privacy scores in Table 2 almost have the same trend. And we can see privacy scores provide better quantitative results. In Table 2, we find that the recognition accuracy of Masking+MixUp($k$=2) is better than that of MixUp-FD($k$=2). Note that we train the models using the frequency-domain data. The frequency-domain energy distribution of each image is uneven, and the direct current (DC) partly accounts for more than 90% of the energy Wang et al. (2022b). The training of MixUp-FD($k$=2) uses all frequency-domain components. And Masking+MixUp removes the redundant frequency-domain components (the DC component is also removed) for training the recognition models, making the training data used more evenly distributed, and the accuracy of it may be slightly better.

We follow the powerful assumption for attackers in Section 3, i.e., the attackers have the ability to infer all masked data. To reconstruct the original image from the masked data, we select the same CGAN as in PPFR-FD used as a black-box attack. For the data used in the reconstruction experiments for AdaMasking, we generate hybrid masked data according to the statistical proportion of different $k$ values participating in the training. It is worth noting that under strong assumptions, the attacker can directly obtain the masked data. For cases where Instahide and MixUp are used alone, even without reconstruction attacks, they are hacked by the method in Carlini et al. (2020). And the proposed method has good performance in recognition performance and privacy protection ability.

From Fig. 3, we provide more results from the black-box attack by CGAN for different masking methods, and the proposed method can better resist the black-box attack than others. For the white-box attack, we can use the methods in PPFR-FD Wang et al. (2022b) and Carlini et al. (2020). Since we propose an adaptive hybrid privacy-preserving method, which is essentially a combination of two masked methods, the combined method can provide a stronger privacy protection ability. For the proposed privacy protection algorithm, we first use the method in Carlini et al. (2020) to decouple the combination of different data. According to the theoretical analysis in Carlini et al. (2020), we can obtain the inaccurate single data before MixUp. Then the white-box attack method in PPFR-FD Wang et al. (2022b) is used to recover the original image. According to the white-box attack method

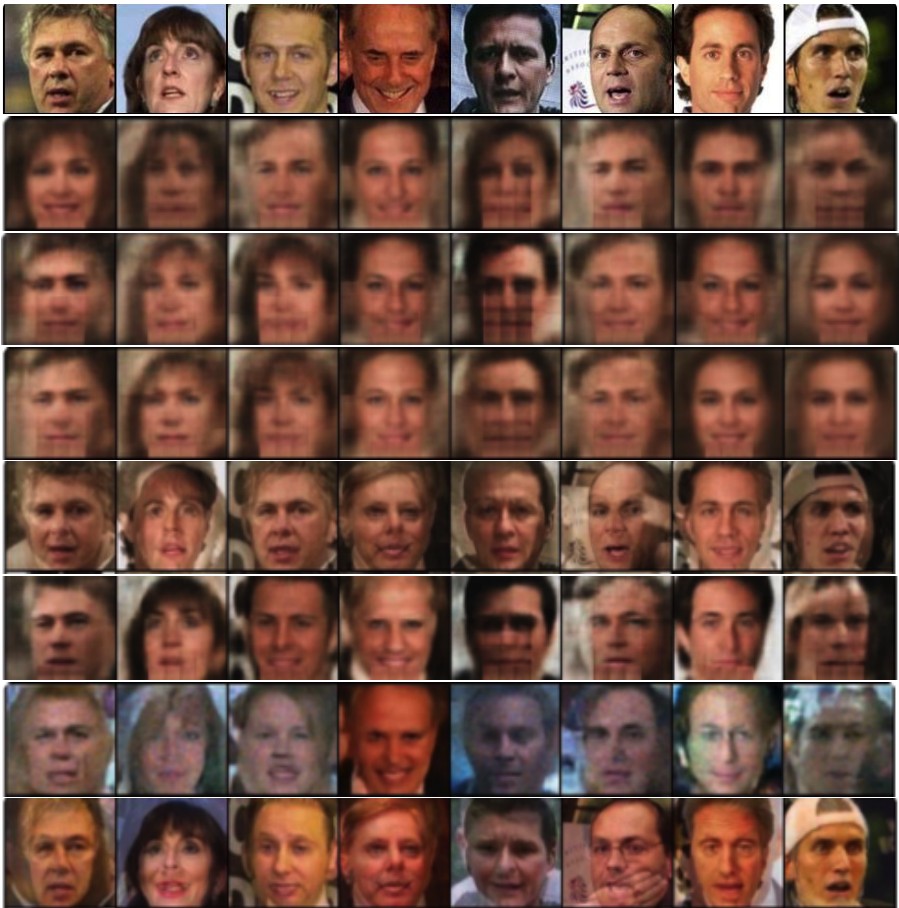

Figure 3: Face images recovered by CGAN for different masking methods. Results from the 1st row to the last row correspond to the original images, Masking+AdaMixUp(k:2-4), Masking+AdaMixUp(k:2-5), Masking+AdaMixUp(k:2-6), MixUp(k=2), PPFR-FD(Masking), DP, and Instahide.

in Wang et al. (2022b), it is difficult to reconstruct the original image even if accurate masked data is used. Note that the masked data available now are inaccurate. Since the introduction of errors, it is more difficult to reconstruct the original image accurately. The above analysis also shows that the proposed algorithm brings stronger privacy protection ability to a certain extent. We also find that, when the size of the candidate set of $k$ is large (e.g., 2-10), the strategy network tends to choose a larger value under the effect of the reward function. A large value $k$ will lead to non-convergence of the FR loss function, and ultimately lead to poor recognition accuracy.

Finally, we give a detailed discussions on the latest masking methods in Appendix.

## 6 CONCLUSION

Aiming at the problem of resisting the face recognition MIAs, this paper proposed an adaptive hybrid masking-based defense algorithm, where the face image is masked by combining frequency-domain masking and adaptive MixUp. The adaptive MixUp carries out the mixing in the frequency domain and was developed based on reinforcement learning. The strategy and face recognition networks can be regarded as two opposing sides in the training process. It can select as many images as possible for mixing to enhance privacy protection while maintaining good recognition accuracy. Experimental results show that the proposed scheme has a better privacy-preserving ability for MIA and recognition accuracy over other existing algorithms. In this work, a set of standard measures for quantifying the effect of masking on privacy protection and face recognition was advocated.

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

# A APPENDIX

| **Algorithm 1: MixUp based ArcFace** |
|---|
| **Input**: feature scale $s$, margin parameter $m$, class number $n$, mixed label list $gt$, mixed label coefficient list $co$, mixed number $k$, full connection layer weight $w_{FC}$, extracted feature $x$ |
| **Output**: class-wise affinity score $metric$ |
| 1:   $x$ = L2Normalization($x$) |
| 2:   $w_{FC}$ = L2Normalization($w_{FC}$) |
| 3:   $fc$ = FullyConnected(data=$x$,weight=$w_{FC}$) |
| 4:   $metric$ = 0 |
| 5:   for $i$ in range($k$): |
| 6:      Original_target_logit = Pick(fc,gt[i]) |
| 7:      $\theta$ = arccos(Original_target_logit) |
| 8:      Marginal_target_logit = $\cos(\theta + m)$ |
| 9:      One_hot = OneHot($gt[i]$) |
| 10:     $metric$ = $metric$+$co[i]$*($fc$+broadcast_mul(One_hot, |
| 11:       expand_dims(Marginal_target_logit - |
| 12:       Original_target_logit,1))) |
| 13:  Return $metric$ = $metric \cdot s$ |

## A.1 MASKING IN THE FREQUENCY DOMAIN

For the completeness of the algorithm description, we give a detailed introduction of PPFR-FD developed in Wang et al. (2022b).

### A.1.1 BLOCK DISCRETE COSINE TRANSFORM (BDCT)

As the first step of PPFR-FD, BDCT is performed on the face image after it has been converted from a color image to a gray one. Similar to the convolution in CNN, BDCT is carried out on image blocks with the size $a \times b$ pixels according to the stride $s$. The $a \times b$ BDCT coefficient matrix is produced for each image block. Here, we set $a = 8$, $b = 8$ and $s = 8$. Every element of the coefficient matrix represents a specific frequency component. We collect the frequency components having the same position in the BDCT coefficient matrix to form a frequency channel.

### A.1.2 FAST FACE IMAGE MASKING

PPFR-FD performs the BDCT and selects channels according to the analysis network (only the DC component is discarded for a high FR accuracy) Wang et al. (2022b). Next, the remaining channels are shuffled two times with a channel mixing in between. After each shuffling operation, channel self-normalization is performed. The result of the second channel self-normalization is the masked face image that will be transmitted to third-party servers for face recognition. The goal of PPFR-FD is to provide a lightweight masking method to make it difficult for attackers to recover the training and inference face images in the FR system. It is also an initial step towards exploring better privacy preservation while maintaining data utility. From the experimental section, we can see that it may be possible to leak some privacy. To enhance its privacy protection capability, we propose a hybrid privacy-preserving policy. We use it as the basis of the proposed method.

## A.2 DISCUSSION ON THE LATEST MASKING METHODS

The first latest masking method Ji et al. (2022) is developed from PPFR-FD Wang et al. (2022b). It proposes a privacy-preserving face recognition method using differential privacy in the frequency domain. Due to the utilization of differential privacy, it offers a guarantee of privacy in theory. This method first converts the original image to the frequency domain and removes the direct component. Then a privacy budget allocation method can be learned based on the loss of the back-end face recognition network within the differential privacy framework. Finally, it adds the corresponding noise to the frequency domain features. Note that compared with PPFR-FD, the method in Ji et al. (2022) does not delete redundant high-frequency channel components. In PPFR-FD, it is pointed out

that the element values in these redundant high-frequency channels are close to 0, and the removal of these components will not significantly affect the recognition accuracy. The privacy protection method based on the learnable differential privacy in Ji et al. (2022) adds most of the noise to these redundant channels with high probability in the learning process. And the noise that is added to identify important channels may be relatively small. If the attacker attacks the masking process and removes the relevant redundant high-frequency channels, most channels important for identification can be obtained. Then it is possible to reconstruct the original image by the black-box reconstruction attack. So, this method has a certain risk of privacy disclosure.

The other latest work is FaceMAE Wang et al. (2022a), which is based on Masked Autoencoders He et al. (2022). It proposes a novel framework FaceMAE, where face privacy and recognition performance are considered simultaneously. Firstly, randomly masked face images are used to train the reconstruction module in FaceMAE. It tailors the instance relation matching (IRM) module to minimize the distribution gap between real faces and FaceMAE reconstructed ones. During the deployment phase, it uses trained FaceMAE to reconstruct images from masked faces of unseen identities without extra training. The masked data are the reconstructed images using trained Face-MAE. And the risk of privacy leakage is measured based on face retrieval between reconstructed and original datasets. Its method of measuring privacy protection capability does not consider the way of using a black-box attack. That is, it is considered to construct data pairs of the reconstructed images using FaceMAE and the original images as the training dataset for the CGAN training. Since the distance consistency loss function is used in the training of FaceMAE, the reconstructed (masked) images using FaceMAE may have some correlation with the original images. This makes it possible for the black-box reconstruction process to reconstruct the original images. For our proposed hybrid privacy protection strategy, FaceMAE can also replace the PPFR-FD method to form a new hybrid privacy protection policy.

From this perspective, the hybrid privacy protection strategy proposed is not only a specific privacy protection strategy but also a privacy protection framework (the basic masked method can be replaced by a better masking algorithm).

