# OpenReview forum: "Hybrid Defense Strategy for Face Recognition Model Inversion Attack"
_ICLR.cc/2024/Conference — ICLR 2024 Conference Withdrawn Submission_

### Official Review · Reviewer_dDsq · 2023-10-13

**Soundness:** 2 fair
**Presentation:** 1 poor
**Contribution:** 2 fair
**Rating:** 1
**Confidence:** 4

**Summary:**

The paper proposes a novel defense mechanism against model inversion attacks (MIAs), which aim to reconstruct private samples from a model's training data. The proposed defense applies MixUp augmentation in the frequency domain with an adaptive number of images that are combined. The individual number of images used is computed by a policy network trained with reinforcement learning. By using an adaptive number of images, the number of images taking part in each masking step can be increased without hurting the model's prediction performance noticeably. The approach is then evaluated against standard MixUp and Masking approaches and promises improvements in both privacy and utility metrics.

**Strengths:**

MixUp training in the frequency domain offers an interesting strategy to defend against common MIAs. Whereas the underlying idea is not novel (e.g., InstaHide), the combination with an adaptive mixing strategy is promising.

**Weaknesses:**

The paper has various flaws and weaknesses.

1.) The paper plagiarizes previous work directly by adopting individual sentences without modification and without appropriate citation. Particularly section 2.1 is taken verbatim from the paper "Plug and Play Attacks" (https://arxiv.org/abs/2201.12179). To demonstrate, here are the sections from this paper under Review and the Plug and Play Attacks Paper:

This paper (Sec. 2.1): _"As we know, most MIAs utilize the ability generative generative adversarial networks (GANs) Goodfellow et al. (2020), and use images as the prior information to generate realistic images. Some MIAs avoided distributional shifts and relied on GANs trained on the same data distribution as the target model Zhang et al. (2020); Chen et al. (2021); Wang et al. (2021), used additional input information such as blurred pictures of a person Zhang et al. (2020), and tailored the attack and its image prior to specific target models Chen et al. (2021); Wang et al. (2021), restricting the reuse and flexibility of the attacks. All approaches focused on low-resolution images, which limits the quality of the extracted features, and have yet to show their applicability for higher resolutions. Struppek
et al. (2022) presents Plug & Play Attacks, which relax the dependency between the target model and image prior, and enable the use of a single GAN to attack a wide range of targets, requiring only minor adjustments to the attack. It also provides a more formal introduction to MIAs and theoretical consideration of ideal MIAs and possible degradation factors."_

Plug and Play Attacks (Abstract): _"[...] we present Plug & Play Attacks, which relax the dependency between the target model and image prior, and enable the use of a single GAN to attack a wide range of targets, requiring only minor adjustments to the attack._

Plug and Play Attacks (Sec. 1): _"For attacking deep neural networks, most MIAs use generative adversarial networks (GANs) (Goodfellow et al., 2014) as image priors to generate realistic images. Previous MIAs [...] avoided distributional shifts and relied on GANs trained on the same data distribution as the target model (Zhang et al., 2020b; Chen et al., 2021; Wang et al., 2021), used additional input information such as blurred pictures of a person (Zhang et al., 2020b), and tailored the attack and its image prior to specific target models (Chen et al., 2021; Wang et al., 2021), restricting the reuse and flexibility of the attacks. Also, all approaches focused on low-resolution images, which limits the quality of the extracted features, and have yet to show their applicability for higher resolutions."_

2.) The paper lacks an appropriate presentation and writing. Insufficient introduction of terms, numerous typos, and badly designed figures make it really hard to follow and understand the paper. Approaches like InstaHide should be introduced more formally. Otherwise, following the introduction section is already hard for people not familiar with this method. Similarly, following the explanation of the proposed approach in Section 4 should be improved.

3.) There are numerous details missing in the paper. For example, Fig. 1 states attack results for the MIA by Zhang et al. Without reading the caption, it is unclear, which image corresponds to which defense mechanism. Also, which dataset has been used, how are the depicted samples selected, and what are the evaluation metrics for the attack? Additionally, in this specific case, I doubt that the MIA by Zhang et al. has been used since the approach is designed for 64x64 image resolutions and the depicted images in Fig. 1 clearly have higher resolutions.

4.) The evaluation is only done insufficiently. It is unclear why the paper first introduces model inversion attacks and then never evaluates the proposed defense mechanism on them but explores an attack scenario in which the adversary has access to the shared gradients in a federated learning setting. The paper has to better place the defense mechanism into the literature and conduct a more thoughtful evaluation. Also, details on the performed attack and the CGAN are lacking.

**Questions:**

- What is the intuition to maximize the loss function of the FR? Why does a higher classification loss lead to better privacy?
- Why is the evaluation not done on the introduced model inversion attacks?
- Should the Feature level evaluation model F1 not be trained on a mix of original and masked data to avoid unexpected behavior for the distribution shift induced by the masking steps?

**Details Of Ethics Concerns:**

Parts of the paper are taken verbatim from another paper (see Weaknesses). An appropriate citation is missing.

---

### Official Review · Reviewer_tfZL · 2023-10-30

**Soundness:** 2 fair
**Presentation:** 1 poor
**Contribution:** 2 fair
**Rating:** 3
**Confidence:** 3

**Summary:**

This paper presents an improved MixUp approach utilizing reinforcement learning to safeguard against membership inference attacks (MIA) while preserving the accuracy of face recognition.

**Strengths:**

Addressing the issue of defending against MIA in the field of face recognition holds significant importance.

The authors provide concrete visual data to demonstrate the effectiveness of their defense strategy.

**Weaknesses:**

The presentation of the material appears to be quite confusing, as indicated in questions section.

The motivation behind their design is unclear. The objective in section (5) is perplexing. It's not evident why there is a need to optimize a strategy network that maximizes the original loss function. This seems counterintuitive, as the defense goal isn't to reduce face recognition accuracy but rather to lower the success rate of image inversion.

The authors allocate approximately one page to discuss the metric, but the definition of the metric is somewhat obscure and potentially introduces bias. For instance, they introduce a comprehensive metric that combines s1, s2, s3, s4 with $\alpha$ and $\beta$, but they arbitrarily set $\alpha=0.4$ and $\beta=0.6$" based on the intuition that "masking images at the visual level is easier to achieve, so we will pay more attention to the indicators at the attack level". This approach lacks objectivity.

The experimental results are not well organized and difficult to comprehend.

Evaluating the primary contribution of this paper is challenging due to its writing style. It is recommended that the authors carefully proofread and revise their paper before the formal submission.

**Questions:**

Presentation issues:
1. In Eq. (1), $x_t$, $x_{b,t}$, $y_{bt}$ are mentioned below the equation, but they are simply not used in the equation.

2. In Eq. (1), the notation $x_{mix}$ is used without prior definition, and $x_t$ is not utilized in the equation.

3. In Table 2, the meaning of the numbers in the last four columns is unclear and should be explained.

4. The reason why solving (2) alone is insufficient for ensuring good defense performance is not adequately explained. The introduction of the two loss terms, L1 and L2, lacks justification.

5. The algorithm workflow for solving (6) and (8) is missing, making it unclear how the authors arrive at their solutions.

---

### Official Review · Reviewer_LCMn · 2023-11-01

**Soundness:** 2 fair
**Presentation:** 2 fair
**Contribution:** 2 fair
**Rating:** 5
**Confidence:** 4

**Summary:**

The paper proposes a hybrid method for defending the model inversion attack on the face recognition model. Technically, a mixup strategy with reinforcement learning technique to adaptively mix several images while maintaining a satisfactory accuracy of recognition. Empirically, the proposed method presents advantages in recognition accuracy and attacking robustness compared with other data augmentation methods.

**Strengths:**

The investigated problem of model inversion attack is novel, important, and under-explored.

The proposed combination of RL and data augmentation for defending MIA is reasonable and justified to be effective.

The presentation and drawn figures are generally clear and easy to understand.

The paper elaborates on several technique details and experimental settings.

Some case studies with visualization are also provided.

**Weaknesses:**

The writing of the paper can be largely improved. The overall writing style is too technical but lacking of essential and principle designs. Besides, Section 3 is too short to be a section.

Besides, the mathematical notations and equations can be improved to be clearer. It would be better to summarize the frequently used notations in one table or sentence.

The paper is empirically driven and lacks in-depth analysis, whether from methodological or theoretical perspectives, for understanding the under-explored problem of MIA.

The technical novelty is neutral, although the proposed method skillfully combines both worlds of reinforcement learning and data augmentation for defense.

The complexity of the method is unknown. Besides, the running-time efficiency of the proposed method is not reported. It would be better to show the training procedure of the RL model, e.g., by drawing the training curves of rewards, loss, or other metrics.
There is a lack of essential information about the used datasets.

Several defense methods for MIA, e.g., [1,2,3], are not included in the paper. Although they may utilize different kinds of techniques for defense, I would suggest at least constructing a more comprehensive related work and a further discussion.

[1] Adversarial Neural Network Inversion via Auxiliary Knowledge Alignment. CCS 2019.

[2] Bilateral Dependency Optimization: Defending Against Model-inversion Attacks. KDD 2022.

[3] On Strengthening and Defending Graph Reconstruction Attack with Markov Chain Approximation. ICML 2023.

**Questions:**

Please refer to the above weakness part.